# Synthesis and Characterization of Block Copolymers of Poly(silylene diethynylbenzen) and Poly(silylene dipropargyl aryl ether)

**DOI:** 10.3390/polym13091511

**Published:** 2021-05-07

**Authors:** Man Gao, Chengyuan Shang, Jixian Li, Gang Han, Junkun Tang, Qiaolong Yuan, Farong Huang

**Affiliations:** 1Key Laboratory of Specially Functional Polymeric Materials and Related Technology (Ministry of Education), School of Materials Science and Engineering, East China University of Science and Technology, Shanghai 200237, China; mangao96@163.com (M.G.); chmxy521@163.com (J.L.); ganghmail@163.com (G.H.); jktang@ecust.edu.cn (J.T.); qlyuan@ecust.edu.cn (Q.Y.); 2Research and Application Center for Structural Composites, Aerospace Research Institute of Materials & Processing Technology, Beijing 100076, China; shangchengyuan@163.com

**Keywords:** poly(silylene diethynylbenzen)–*b*–poly(silylene dipropargyl aryl ether), three-block copolymer, silicon-containing arylacetylene resin, high-performance resin matrix for composites

## Abstract

Poly(silylene diethynylbenzene)–*b*–poly(silylene dipropargyloxy diphenyl propane) copolymer (ABA-A), poly(silylene diethynylbenzene)–*b*–poly(silylene dipropargyloxy diphenyl ether) copolymer (ABA-O), and a contrast poly(silylene diethynylbenzene) with equivalent polymerization degree were synthesized through Grignard reactions. The structures and properties of the copolymers were investigated via hydrogen nuclear magnetic resonance, Fourier transform infrared spectroscopy, Haake torque rheometer, differential scanning calorimetry, dynamic mechanical analysis, thermogravimetric analysis and mechanical tests. The results show that the block copolymers possess comprehensive properties, especially good processability and good mechanical properties. The processing windows of these copolymers are wider than 58 °C. The flexural strength of the cured ABA-A copolymer reaches as high as 40.2 MPa. The degradation temperatures at 5% weight loss (*T*_d5_) of the cured copolymers in nitrogen are all above 560 °C.

## 1. Introduction

Silicon-containing arylacetylene resins, a kind of new high-performance resin, have shown potential applications in the fields of aerospace navigation, military industry and electronic information [1]. Many researchers have studied silicon-containing arylacetylene resins. In 1967, Luneva et al. [2] introduced elemental silicon into polymers with acetylene groups to prepare a kind of polymer with good heat resistance. The prepared polymers begin to decompose at above 500 °C. In 1990, Corriu et al. [3,4,5,6] prepared a silicon-containing arylacetylene polymer by means of a Sonogashira reaction. In 1994, Itoh et al. [7,8,9] reported poly[(phenylsilylene)-ethynylene-1,3-phenyleneethynylene] (MSP) via dehydrogenative coupling reaction under the catalysis of magnesium. The cured MSP polymer shows extremely high thermal stability with a degradation temperature at 5% weight loss of 860 °C and a residue yield at 1000 °C of 94%. In 2001, Buvat’s group [10] synthesized a phenylacetylene-terminated poly(silylene ethynylene phenylene ethynylene) (BLJ) with good processability. Since 2002, our research group has developed many new silicon-containing arylacetylene resins, which are called poly(silylene arylacetylene)s (PSAs) [11]. In 2004, Yan et al. [12] reported phenyl acetylene-terminated PSA polymers by condensation of dimethyldichlorosilane and organic magnesium Grignard reagents of diethynylbenzene and phenylacetylene. The cured polymers decomposed at a temperature higher than 460 °C and the decomposition of residue yield at 800 °C was higher than 80%, which indicated that the polymers had high heat resistance. To date, a series of PSA polymers have been studied [13,14,15,16,17,18,19]. However, due to the rigidity of the aromatic ring structure and high content of reactive acetylene groups in PSA polymers, the cured PSA polymers are somewhat brittle. In addition, the cost of PSA polymers is high. These points limit the wide application of PSA polymers.

The preparation of arylacetylene monomers requires multistep reactions, and the process is complicated and costly. Monomers containing dipropargyl ether groups can be synthesized with high yield and low cost, which has attracted people’s attention [20]. Furthermore, poly(silylene dipropargyl aryl ether) has good solubility, processabilities, mechanical properties and thermal stabilities [21,22,23]. By introducing the dipropargyl ether units into PSA polymer chains, the produced polymer has more flexible chains. As a result, the processability of the produced polymer would be better and the cost would be reduced at the same time. The comprehensive properties of the polymer would be improved without excessively sacrificing the heat resistance. In 2014, Wang et al. [24] synthesized a series of block copolymers of poly(siloxane arylacetylene) and poly(silylene arylacetylene) (SiO-*b*-PSA). The copolymers showed good heat resistance and toughness. In 2018, Yang et al. [25] prepared poly(silylene dipropargyl aryl ether) (SAPE-BA, SAPE-DPE) through Grignard reactions, and used them to modify PSA polymers. The modified PSA polymers possessed better fluidity, processability and mechanical properties.

In order to improve the properties of a PSA polymer and decrease the production cost, block copolymers with new structures were designed and synthesized. On one hand, we use some dipropargyl aryl ether instead of part of diethynylbenzene to synthesize the copolymers to reduce the cost. On the other hand, different structural dipropargyl aryl ether units were introduced into the copolymer to investigate the effect of the structures on the properties of the copolymer. Finally, comprehensive properties of the copolymer could be obtained without excessively sacrificing the heat resistance. In this study, two block copolymers of poly(silylene diethynylbenzen) and poly(silylene dipropargyl aryl ether) were prepared through Grignard reactions. The structures and properties of the block copolymers were characterized in detail.

## 2. Experimental Methods

### 2.1. Materials

Dichlorodimethylsilane was supplied by Shanghai Macklin Biochemical Co., Ltd., Shanghai, China. 1,3-Diethynylbenzene (*m*-DEB), tetrahydrofuran (THF), anhydrous sodium sulfate (Na_2_SO_4_), magnesium powder, toluene and ethyl bromide were supplied by Shanghai Titan Scientific Co., Ltd., Shanghai, China. Acetic acid and concentrated hydrochloric acid were bought from Shanghai Sinopharm Chemical Reagent Co., Ltd., Shanghai, China. All chemicals were of analytical reagent (AR) grade and used as received. Dipropargyloxy diphenyl propane and dipropargyloxy diphenyl ether were prepared from bisphenol A and dihydroxydiphenyl ether according to the literature [26,27].

### 2.2. Synthesis of Block Copolymers

The polymers were synthesized through Grignard reactions under dry nitrogen, as shown in Scheme 1. The reaction device was a four-necked round-bottomed flask equipped with a mechanical stirrer, a condenser, a constant-pressure dropping funnel and a thermometer. The synthesis of block polymers was carried out simultaneously in two sets of devices with different sizes, and the specific synthesis steps of the polymers are as follows.

Device 1: Magnesium powder (6.66 g, 0.278 mol) and THF (75 mL) were charged in a 500 mL flask. Ethyl bromide (27.50 g, 0.252 mol) in THF (15 mL) was added dropwise when the reaction temperature was below 30 °C. Thereafter, the mixture was heated to 40 °C and stirred for 1 h to produce ethyl magnesium bromide. The reaction mixture was cooled to 20 °C, and a solution of *m*-DEB (15.14 g, 0.120 mol) in THF (75 mL) was added slowly. The reaction mixture was heated to 70 °C right after and the reaction was then continued for 1.5 h at 70 °C. After the reaction product was cooled to 20 °C with an ice water bath, dichlorodimethylsilane (10.33 g, 0.080 mol) in THF (15 mL) was slowly added dropwise. Afterwards, the reaction solution was heated to 70 °C, and refluxed for 1.5 h to obtain Grignard macromonomer reagent A.

Device 2: At the same time, to another 250 mL flask, magnesium powder (2.22 g, 0.093 mol) and THF (25 mL) were added. Ethyl bromide (9.20 g, 0.084 mol) in THF (10 mL) was added dropwise when the reaction temperature was below 30 °C. Thereafter, the mixture was heated to 40 °C and stirred for 1 h to produce ethyl magnesium bromide. The reagents were cooled to 20 °C, and a solution of dipropargyl aryl ether (0.040 mol) in THF (35 mL) was added into the flask. Then, the reaction mixture was heated to 70 °C and the reaction was continued for 1.5 h at 70 °C. After the temperature of the reaction product dropped to 20 °C with an ice water bath, dichlorodimethylsilane (7.75 g, 0.060 mol) in THF (10 mL) was slowly added dropwise. Afterwards, the reaction solution was heated to 70 °C, and refluxed for 1.5 h to obtain dichlorodimethysilyl-terminated macromonomer B solution.

Finally, B solution was slowly added dropwise to the reaction reagent A in Device 1. Then, the mixture in the flask of Device 1 was heated to 70 °C and refluxed for 2 h. Afterwards, the mixture temperature was cooled to below 30 °C with an ice water bath, and a mixed solution of toluene (125 mL) and glacial acetic acid (20.00 g, 0.320 mol) was added into the flask. Then, 5% hydrochloric acid solution (105 mL) was added dropwise. The obtained polymer solution was placed in a 1000 mL separatory funnel, and washed with deionized water till the washing water became neutral. The organic solution separated was dried with Na_2_SO_4_ overnight. After suction filtration, distillation and vacuum drying, an ABA block copolymer was obtained as a yellow solid with a yield of 91–98%.

When dipropargyl aryl ether is dipropargyloxy diphenyl propane, the obtained block copolymer is named poly(silylene diethynylbenzene)–*b*–poly(silylene dipropargyloxy diphenyl propane) block copolymer (coded name ABA-A). When dipropargyl aryl ether used is dipropargyloxy diphenyl ether, the copolymer obtained is named poly(silylene diethynylbenzene)–*b*–poly(silylene dipropargyloxy diphenyl ether) block copolymer (coded name ABA-O). As a comparison, a poly(silylene diethynylbenzene) with an equivalent polymerization degree was synthesized from *m*-DEB and dichlorodimethylsilane (DCDMS) with molar ratio [*m*-DEB]/[DCDMS] of 8/7 and was coded as AAA.

### 2.3. Characterizations

A hydrogen nuclear magnetic resonance (^1^H-NMR) spectrum was obtained on an Avance 500 MHz spectrometer (Bruker, Billerica, MA, USA) with tetramethylsilane as an internal standard. A polymer (2–10 mg) dissolved in CDCl_3_ solvent (0.6 mL) and the polymer solution were tested. Fourier transform infrared spectroscopy (FT-IR) was performed on a Nicolet iS10 infrared spectrometer (Thermo Scientific, Madision, WI, USA) with a scanning range of 400–4000 cm^−1^ and a resolution of 0.09 cm^−1^, the number of scans was 32 and the sample was prepared by a KBr plate. X-ray diffraction patterns were recorded at room temperature by monitoring the diffraction angle 2θ from 10° to 80° on a D/Max 2550 VB/PC rotating anode X-ray powder diffractometer (Rigaku Corporation, Tokyo, Japan), and the scanning speed was 0.001°/s. Gel permeation chromatography (GPC) was carried out on a Waters 1515 chromatography instrument using tetrahydrofuran as an eluent, and the flow rate was 1 mL/min. Rheological behavior was determined on a Thermo Haake RS600 rheometer system (Thermo Electron Corporation, Karlsruhe, Germany) in the range of 80–200 °C, and the shear rate and heating rate for the viscosity measurements were 0.01 s^−1^ and 2 °C/min, respectively. Differential scanning calorimetry (DSC) analysis was carried out at a heating rate of 10 °C/min and a flow rate of 50 mL/min of nitrogen on a TA Q2000 analyzer (TA, New Castle, DE, USA). Thermal gravimetric analyses (TGA) were conducted on a thermogravimetric analysis TGA/DSC 1LF analyzer (METTLER TOLEDO, Greifensee, Switzerland) at a heating rate of 10 °C/min from 40 °C to 900 °C in nitrogen and air atmosphere with a flow rate of 50 mL/min, and the sample amount was 8-10 mg. Dynamic mechanical analysis (DMA) of the cured polymers was conducted on a DMA 1 (METTLER TOLEDO, Greifensee, Switzerland) at a heating rate of 3 °C/min from 35 °C to 450 °C with an oscillation frequency of 1 Hz, and the testing mode used was three-point bending. The mechanical properties were tested by an electronic universal testing machine (SANS CMT 4204, Shanghai, China) according to ASTM D790-17. An average of at least 5 tests were done for each sample. The melting ranges of the polymers, which are the temperature ranges from the beginning of the melting of the polymers to complete melting, were tested with SGW X-4 microscopic melting point apparatus. A polymer (0.2 g) was placed in a solvent (5 mL) and the solubility was determined by observing whether the polymer was soluble or not [28].

### 2.4. Preparation of Cured Copolymers

The release agent was sprayed on the mold before preheating in an oven at 150 °C for 1 h. AAA, ABA-A and ABA-O polymers were weighed and put into the mold. The mold with the polymers was put in a vacuum oven at 150 °C for about 1 h to remove the residual solvent in polymers. Thereafter, the mold with the polymers was moved to another oven for curing. The curing procedure of ABA-A and ABA-O copolymers was: 200 °C/2 h + 230 °C/2 h + 260 °C/2 h + 300 °C/4 h. The curing procedure of AAA polymer was: 170 °C/2 h + 210 °C/2 h + 250 °C/4 h. After the curing was completed, the dark brown cured polymers were released from the mold. The specimen size for flexural tests was 80 × 15 × 4 mm^3^, and the specimen size for DMA tests was 35 × 6 × 2 mm^3^.

## 3. Results and Discussion

### 3.1. Structural Characterization of Copolymers

#### 3.1.1. ^1^H-NMR Analysis

Figure 1 shows the ^1^H-NMR (CDCl_3_) spectra of AAA, ABA-A and ABA-O polymers and the integral areas of various resonance peaks of protons. The peak at 3.08 ppm (d) can be attributed to the proton on the external acetylene group –C≡CH [29]. The peaks at 6.86–7.68 ppm can be attributed to the aromatic protons. For AAA (Figure 1a), the peak at 0.46 ppm (b) is caused by the methyl protons on silicon atoms. The ratio of the integral area at peak (d) and peak (b) 1.00:21.32 is near the designed one of 1.00:21.00. For ABA-A and ABA-O (Figure 1b,c), the peaks at 0.32 ppm (a) and 0.46 ppm (b) are, respectively, attributed to the methyl protons on silicon atoms, and the peaks at 4.63 ppm (e) are caused by methylene protons. As for ABA-A, the peak at 1.62 ppm (c) belongs to the methyl protons on carbon atoms. The ratio of the integral area at peak (a), peak (b), peak (c), peak (d) and peak (e) 6.28:12.83:6.20:1.00:4.12 is near the designed one of 6.00:12.00:6.00:1.00:4.00. In addition, for ABA-O, the ratio of the integral area at peak (a), peak (b), peak (d) and peak (e) 6.45:12.96:1.00:4.31 is near that of 6.00:12.00:1.00:4.00. The proton of deuterated chloroform (solvent) resonates at 7.26 ppm. Therefore, ^1^H-NMR spectra confirm the structures of the polymers.

#### 3.1.2. FT-IR Analysis

Figure 2 shows the FT-IR spectra of the AAA, ABA-A and ABA-O polymers. The absorption at 3286 cm^−1^ is assigned to the stretching vibration of the group ≡C–H [30]. The absorption at 3056 cm^−1^ belongs to the vibration of Ar–H. The characteristic peak at 2154 cm^−1^ is attributed to the stretching vibration of the –C≡C– bond. The absorptions at 1589 and 1508 cm^−1^ are assigned to the C=C bond vibration of the benzene ring. There is a characteristic absorption peak at 1252 cm^−1^ for the symmetric deformation vibration of the Si–CH_3_ bond. For ABA-A and ABA-O copolymers, the absorptions at 2900 and 2864 cm^−1^ belong to stretching vibrations of the –CH_2_– group. The absorption at 1040 cm^−1^ is attributed to the vibration of the Ar–O–C group bond. As for the ABA-A copolymer, the characteristic peak at 2964 cm^−1^ is due to the asymmetric stretching vibration of the –CH_3_ group. According to the results of FT-IR spectra and ^1^H-NMR spectra, the chemical structures of prepared polymers are as expected.

#### 3.1.3. Molecular Weight

The molecular weights of the three polymers were measured by GPC, and are shown in Table 1. The GPC chromatograms of three polymers are shown in the Supporting Information (Appendix A). The corresponding molecular weights (designed) were calculated from the molar ratio of raw chemicals loaded during the synthesis of the polymers, which are shown as the designed value in Table 1. Simultaneously, by analyzing the integral area of peak (b), peak (d) in ^1^H-NMR spectra of the AAA polymer, and the integral area of peak (b), peak (d), peak (e) in ^1^H-NMR spectra of the ABA-A, ABA-O copolymers, the molecular weights of three polymers can be calculated and are listed in Table 1. As shown in the table, the measured molecular weights of the three polymers show little deviation from the designed molecular weights. In addition, it is notable that there is an error between the molecular weights measured by GPC and the molecular weights calculated. The reason is probably that the internal standard in the GPC test is polystyrene (PS), and the chemical structures and molecular weights of AAA, ABA-A and ABA-O polymers are largely different from those of PS [31].

#### 3.1.4. X-ray Diffraction

The XRD diagrams of the three polymers are shown in Figure 3. The XRD curves of the three polymers are similar, with two big but not sharp diffraction peaks. This illustrates that the three polymers are all partially crystalline. Moreover, the peak intensities of the AAA, ABA-A and ABA-O polymers were successively weakened. It is probable that the introduction of dipropargyl ether in the molecular chains decreases the rigidity of molecular chains. As a result, the crystallinity of the polymers decreases.

### 3.2. Processabilities of Copolymers

Table 2 shows the solubility of the AAA, ABA-A and ABA-O polymers tested at room temperature, and typical photos for the dissolution states of the three polymers in several solvents are shown in Figure 4. The AAA, ABA-A and ABA-O polymers can be completely dissolved in most common solvents, such as tetrahydrofuran, toluene and dichloromethane. However, AAA, ABA-A and ABA-O polymers are insoluble in methanol, petroleum ether and ethanol. Generally speaking, resins with good solubility could expand their application fields, especially in the field of advanced composite materials.

The melting temperature ranges of the AAA, ABA-A and ABA-O polymers were measured with a microscopic melting point apparatus. The results are shown in Table 3. It can be seen that the melting temperature ranges of the three polymers are all wide, and the melting temperature of the AAA, ABA-A and ABA-O polymers decreases successively. This can be explained by the fact that the three polymers are all partially crystalline. As the temperature increases, the amorphous part melts first, and the crystalline part does not melt initially. As the temperature rises further, the crystalline part begins to gradually melt [32]. The flexibility of the main chain of AAA, ABA-A and ABA-O polymers increases, and thereby the melting entropy Δ*S*_m_ of the polymers also rises successively. According to the thermodynamic calculation formula for the melting temperature of a crystalline polymer, *T*_m_ = Δ*H*_m_/Δ*S*_m_, the increase in Δ*S*_m_ will make the melting temperature of the polymer decrease.

Figure 5 shows the viscosity–temperature curves of the three polymers. As the temperature increases, the viscosity of the polymers shows an obvious initial fall. Then, the viscosity is stable and finally goes up clearly at around the curing temperature of polymers. As shown in Table 3, the processing windows of AAA, ABA-A and ABA-O polymers are 135–181 °C, 132–190 °C and 117–182 °C, respectively. The processing window width in the low-viscosity range for AAA, ABA-A and ABA-O polymers successively increases from 46 to 65 °C. In addition, it can be seen that the temperature when the viscosity of the AAA, ABA-A and ABA-O polymers begins to drop decreases in sequence, indicating that the melting temperature of the three polymers decreases successively, which is consistent with the change tendency for the melting temperature range of the three polymers. Generally speaking, the ABA-A and ABA-O copolymers possess wide processing windows and display better processability as compared with the AAA polymer.

### 3.3. Thermal Curing Reactions of Copolymers

Figure 6 shows the DSC curves of the three polymers. As shown in Figure 6, the endothermic peak of AAA, ABA-A and ABA-O polymers is ascribed to the melting of polymers, and all the exothermic peaks are assigned to the crosslinking reactions of the polymers [33]. Moreover, the AAA polymer has only one curing exothermic peak, while ABA-A and ABA-O copolymers possess one curing exothermic peak with a shoulder, and the exothermic peak moves to higher temperature region. There are at least two reactions during the curing, including the cyclotrimerization of the –C≡C– group and the Diels–Alder reaction of ethynyl phenyl and the –C≡C– group. The specific initial curing temperature, peak temperature, termination temperature and exothermic enthalpy data are shown in Table 4.

Furthermore, the curing reaction kinetics of the three polymers were investigated via non-isothermal DSC experiments. DSC curves for the ABA-A copolymer at five different heating rates are shown in Figure 6. The kinetic parameters are determined by both the Kissinger method and Ozawa method [34,35]. The corresponding equations are shown in Equations (1) and (2), respectively:(1)$$\ln\left(\frac{\beta }{T_{P}^{2}}\right)=\ln\left(\frac{AR}{E_{a}}\right)-\frac{E_{a}}{R}\cdot \frac{1}{T_{p}}$$
(2)$$\ln\left(\beta \right)=C-1.052\frac{E_{a}}{RT_{p}}$$
where *A* is the prefactor, *T_p_* the curing peak temperature, *E_a_* the apparent reaction activation energy, *R* the gas constant, C the constant and *β* the heating rate. ln(*β*/*T_p_*^2^) versus 1/*T_p_* and ln*β* versus 1/*T_p_* for the ABA-A copolymer are separately plotted and fitted, as shown in Figure 7. The plots for AAA and ABA-O polymers are shown in the Supporting Information (Appendix A). Table 5 shows the *E_a_* for the curing reaction of the polymers, which is calculated by the slope of the lines. As shown in Table 5, the activation energy of the AAA polymer is 99.4 kJ/mol, while the activation energies of the ABA-A copolymer and ABA-O copolymer are 147.5 and 165.7 kJ/mol, respectively. The apparent curing reaction activation energies obtained by the Kissinger method and Ozawa method for ABA-A and ABA-O copolymers are higher than that of the AAA polymer. This indicates that the curing reactions of the AAA polymer take place more easily than those of ABA-A and ABA-O copolymers. The main reason could be that the thermal reaction activity of the propargyl group is higher than that of the acetylene group.

### 3.4. The Properties of Cured Copolymers

#### 3.4.1. Mechanical Properties of Cured Copolymers

Table 6 shows the mechanical properties of the cured polymers at room temperature. As shown in the table, the flexural strength and flexural modulus of cured ABA-A and ABA-O copolymers are improved compared to the cured AAA polymer. This may be due to the introduction of dipropargyl ether in the molecular chains, leading to increases in molecular polarity, and increases in the intermolecular interaction force between the molecular chains. The flexural strength and flexural modulus of cured the ABA-A copolymer reach 40.2 MPa and 3.1 GPa, respectively. The cured ABA-A and ABA-O copolymers possess good mechanical properties.

#### 3.4.2. Thermal Properties of Cured Copolymers

The thermal mechanical properties of cured AAA, ABA-A and ABA-O polymers were assessed by DMA in the temperature range of 50 °C to 450 °C under nitrogen. Figure 8 shows the DMA curves of cured AAA, ABA-A and ABA-O polymers. As shown in Figure 8, there was no obvious change of the storage modulus and loss tangent (tan δ) of the three polymers. This illustrates that the polymers have no glass transition in the temperature range. Thus, the cured polymers display excellent heat resistance.

The thermal stability of cured polymers was assessed by TGA up to 900 °C under nitrogen and air. The TGA analysis results of the cured polymers are listed in Table 7. As shown in the table, the three cured polymers possess excellent thermal stabilities. The degradation temperatures at 5% weight loss (*T*_d5_) of the cured AAA, ABA-A and ABA-O polymers in nitrogen atmosphere are all higher than 560 °C, and residue yields at 800 °C are higher than 87.2%. It can be seen that the thermal stabilities of the cured ABA-A and ABA-O copolymers decrease as compared with those of the cured AAA polymer. This probably results from the ether and iso-propylene groups in these polymers being easy to break at high temperature. Moreover, the *T*_d5_ for the cured ABA-O copolymer in nitrogen atmosphere is higher than that for the cured ABA-A copolymer, showing that the diphenyl ether structure is more stable than diphenyl propane. In addition, the *T*_d5_ values of the three cured polymers in air atmosphere are all above 518 °C, and the residue yields at 800 °C are all above 24.3%. It is noted that *T*_d5_ for the cured AAA polymer is close to the *T*_d5_ for cured ABA-A and ABA-O copolymers. This indicates that the copolymers possess good thermoxidative stability. Furthermore, there are differences of *T*_d5_ in nitrogen and in air for the cured polymers. Generally speaking, *T*_d5_ in nitrogen is higher than that in air. This is due to the fact that there are oxidation reactions of the polymers in air, while there are no oxidation reactions of the polymers in nitrogen.

## 4. Conclusions

In this paper, the block copolymers of poly(silylene diethynylbenzen) and poly(silylene dipropargyl aryl ether) through multistep reactions are successfully synthesized. The results demonstrate that the block copolymers have good processability with a processing window wider than 58 °C. The cured ABA-A and ABA-O copolymers show good mechanical properties compared to the cured AAA polymer. The flexural strength of the cured ABA-A copolymer reaches 40.2 MPa. The cured ABA-A and ABA-O copolymers display excellent heat resistance. There is no glass transition observed in the temperature range of 50 to 450 °C. The *T*_d5_ for the cured ABA-O copolymer in nitrogen is higher than that for the cured ABA-A copolymer. Moreover, the *T*_d5_ and *Y*_800°C_ in nitrogen of all cured copolymers are above 560 °C and over 87.2%, respectively. In all, the ABA-A copolymer has better potential for further practical applications, especially in the field of resin matrix composite materials. In addition, the block copolymerization of different alkyne monomers is a good approach to control the properties of PSA resins.

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
