# Peer review of "Synthesis and Characterization of Block Copolymers of Poly(silylene diethynylbenzen) and Poly(silylene dipropargyl aryl ether)"

_polymers, 2021, doi:10.3390/polym13091511_

Round 1
Reviewer 1 Report
The manuscript reports Synthesis and characterization of block copolymers of poly(si-2lylene aryl acetylene) and poly(silylene dipropargyl aryl ether), and there are several points that need to be corrected of clarify:
-In introduction section, It would be interesting if refer some works of using Grignard reaction for obtaning silicates polymers, due the Introduction is poor in background.
-I consider that characterizations must be placed after the Synthesis of block copolymers. In reference of how characterizations were carried out, it is needed to provide more details, for instance, in NMR, which was the polymer concentration? IN FTIR, how was done the analysis? I mean which technique was used: KBr plate, ATR, casting; how many scan were done? For XRD, which range was used, scanning speed? For GPC, what kind of column type was used, solved flow, etc.
For thermal analysis, which was the temperature range, sample amount, environment (oxidative or inert), heating rate, in DMA which kind of clamp was used? For mechanical tests, how many replies were carried out? Authors indicate, melting ranges, did it means melt flow?
-In line 153, correct, "blongs".
-All the figures need to be enlarged due the data in them are in some cases negligible.
-Please refer works that report similar signals for FTIR and NMR, due only describe the peaks in spectra.
-Lines 186-187, it is not clear what is mean that use of PS for calibration in GPC is the reason of differences in MW values designed and calculated by means GPC, and the obtained values are considered lows to be considered polymer or they are in range of oligomers?
-Which was the base for the selection of solubility test?
-Line 242, which two reactions corresponds the peaks observed in DSC according with scheme 1?
-What is it mean that curing reactions are easier to take place for polymer AAA? And why polymers ABA-A and ABA-O present that behavior?
-What is the reason that cured polymers ABA-A-, ABA-O have higher mechanical properties than cured AAA?
-For fig 7, I recommend to zoom the Tan delta range, from 0 to 0.15 in the aim to identify possible polymers transitions, due in the way that is plotted it is not possible to see clearly, and why decide that for E' use a range from -2 to 5? If the values are in a range from 3 to 4, and which units corresponds to this axis?
-I recommend that thermal properties obtained from TGA decide to present only in fig 8 or table 7, due data are duplicated in the way that are presented
-Why polymer AAA has better thermal stability compared with ABA-A, ABA-O copolymers?
-Line 311, in conclusions, ABA-A and ABA-O have better mechanical properties compared with who?
In general the manuscript has a lack of scientific discussion of results, the authors need to describe the "why" of obtained results as was indicated in previous comments.
Author Response
We appreciate very much for your insightful comments. Those comments are all valuable and very helpful for revising and improving our paper. We have studied comments carefully and have made correction which we hope meet with your approval. The responses to your comments can be found in the word file.

Reviewer 2 Report
The paper submitted by Gao et al. deals with the synthesis and physicochemical characterization of three copolymer samples based on poly(silylene arylacetylene) and poly(silylene dipropargyl aryl ether) having different middle groups.
The paper is clear, well written and the conclusions are supported by the results. However, some corrections are needed in order to increase the overall quality of the manuscript before the publication:
- In the introduction section the authors must explain why they have synthesized three copolymer samples (which was the scope in studying the three samples?!)
- Please increase the size of the chemical formulas provided in fig 1
- Please add the units in table 1 for molar masses (g/mol). Replace polydispersity with Đ and provide the GPC chromatograms. Also, indicate which peaks were takin into account in order to calculate the Mn by NMR.
- For the sake of clarity, please add some photos with insoluble, partial soluble and insoluble samples in different solvent (section 3.2)
- Please explain why such a difference was noticed between the thermal properties in N2 compared to air (fig 8 and table 7).
- In the conclusion section the authors must indicate which is the better copolymer sample for further practical applications.
Author Response

(The authors gave the same response as above.)

Round 2
Reviewer 1 Report
The manuscript shows an improvement compared with previous version, and the observations were corrected, so after this I consider that manuscript can be consider to be publish.